# LncRNA NEAT1 protects uremic toxin-induced intestinal epithelial barrier injury by regulating miR-122-5p/Occludin axis

Meng Han[1☉], Pathuama P[1☉], Jinhai Tian[2], Chen Wang[1], Shengnan Zhou[1], Lina Fu[1], Libin Wang[3]*, Na Tian[1]*

1 Department of Nephrology, General Hospital of Ningxia Medical University, Ningxia, China, 2 The Biochip Research Center, General Hospital of Ningxia Medical University, Ningxia, China, 3 Huazhong University of Science and Technology Union Shenzhen Hospital/ Shenzhen Nanshan Hospital, Guangdong, China

☉ These authors contributed equally to this work.
* tianna@nxmu.edu.cn (NT); wanglibin007@126.com (LW)

## Abstract

### Background

Long non-coding RNA(LncRNA) has been reported to be associated with intestinal barrier damage. The aim of this study was to explore the mechanism of lncRNA Nuclear enriched abundant transcript 1 (NEAT1) in uremic toxin-induced intestinal epithelial barrier injury.

### Methods

Human colon cancer cells (Caco-2) were used to establish intestinal epithelial injury models with the urea treatment in different conditions. Cell Counting Kit-8 (CCK-8) and Western blot screening the best concentration and time. The expressions of lncRNA NEAT1 and miR-122-5p were measured by quantitative real-time polymerase chain reaction (qRT-PCR). Western blot and immunofluorescence were used to detect the expression of tight junction proteins Occludin, ZO-1 and Claudin-1. Sodium fluorescein was used to detect the paracellular permeability of intestinal epithelial injury models. The binding of miR-122-5p to lncRNA NEAT1 and Occludin was determined by bioinformatics analysis and dual luciferase reporter assay.

### Results

The best condition for the injury model was urea treatment in 144 mg/dl for 48 hours. With the increase of urea intervention time and concentration, the damage degree of intestinal epithelial cells is aggravated. Based on the qRT-PCR results, lncRNA NEAT1 was significantly down-regulated in the model group. Meanwhile, the tight junction proteins Occludin, ZO-1 and Claudin-1 were significantly reduced. The

**Data availability statement:** All relevant data are within the manuscript and its Supporting Information files.

**Funding:** This project is funded by the Natural Science Foundation of China (NSFC) (No. 81960144, 82360153), Ningxia Natural Science Foundation (Key Project, 2022AAC02062), Key R&D Projects in Ningxia Autonomous Region (2022BEG03120);Tian Na received the award. The website of Natural Science Foundation of China and Ningxia Natural Science Foundation were listed as follow: 'https://isisn.nsfc.gov.cn/' and 'https://gl.nxinfo.org.cn/lib/index.html#/page/home'. The funder had no role in study design,data collection and analysis, decision to publish, or preparation of the manuscript.

**Competing interests:** The authors have declared that no competing interests exist.

permeability of sodium fluorescein was significantly increased in the model group. Overexpression of lncRNA NEAT1 can alleviate the above performances. As the target gene of lncRNA NEAT1, miR-122-5p is significantly up-regulated in the model group. The dual luciferase reporter assay proved that miR-122-5p was targets to Occludin. The protective effect of overexpression lncRNA NEAT1 on intestinal epithelial barrier function is reversed by miR-122-5p mimics.

## Conclusion

LncRNA NEAT1 protects uremic toxin-induced intestinal epithelial barrier injury by regulating miR-122-5p/Occludin axis.

---

## Introduction

Chronic kidney disease (CKD) is a major public health problem that threatens human health. Globally, the prevalence of CKD is approximately 8% to 14%, about 700–840 million of the population being affecting [1,2]. Long-term persistent progression of CKD will enter the End stage renal disease (ESRD), which is uremia. At present, there is no effective treatment for urrmia. Dialysis or kidney transplantation therapy is the main treatment measures. As a result, there is a heavy economic burden on the global economy and healthcare.

   Recent studies have found that the intestinal barrier of patients with CKD exsits obvious dysfunction, and then cooperate with intestinal bacteria and endotoxin into the blood,induce inflammatory response, stimulate the immune system, and cause oxidative stress and systemic inflammation, which are important factors that aggravate the development of CKD and various complications such as cardiovascular disease, malnutrition, and osteoporosis, which directly or indirectly affect the quality of life and prognosis of CKD patients.[3–6]. Therefore, the strategy of "treating the kidney from the intestine" is one of the key targets for delaying the progression of CKD. As a mechanical barrier, the intestinal mucosal barrier is an important line of defense to prevent intestinal bacteria, endotoxins and antigens from entering the blood and lymph. Tight junction (TJ) is the most important component of the intestinal mechanical barrier [7]. TJ is composed of several transmembrane proteins and intracellular scaffold proteins, including Claudin-1, Occludin, zonula occludens-1 (ZO-1). If the integrity of tight junction protein (TJP) is disrupted, the intestinal barrier would damage and the intestinal permeability would increase. This situation can be improved after repaired of the TJP [8,9]. Some studies have initially explored that some uremic toxins such as urea[10], indoxyl sulfate[11] and homocysteine[12] can destroy intestinal TJP and intestinal barrier function. However, the molecular mechanism of these uremic toxins in intestinal epithelial barrier damage is not fully elucidated.

   LncRNA is a group of RNA with more than 200 nucleotides length and no ability to encode proteins [13]. It exerts its regulatory function through various mechanisms, such as binding to proteins as bait, acting as a sponge for miRNA, and acting as a scaffold or guide to regulate the interaction between proteins and genes [14].

LncRNA is also involved in various pathophysiological processes, such as inflammation, apoptosis and oxidative stress [15,16]. Recent studies have shown that lncRNA regulate the expression of TJPs in a variety of diseases. For example, in inflammatory bowel disease, PlncRNA1 overexpression protects intestinal barrier function by regulating the expression of Myc-associated zinc finger protein (MAZ), ZO-1 and Occludin [17]. Another research reported that overexpression of lncRNA MALAT1 protected the integrity of vascular endothelial cells by up-regulating TJPs ZO-1 and Occludin [18]. In diabetic rats, lncRNA H19 deficiency significantly attenuated the damage of endothelial structure by upregulating the expression of junction proteins ZO-1 and Occludin, glycolcalyx protein Syndecan-1, and endothelial activation marker sVCAM-1 and sICAM-1 in diabetic rats[19]. However, the role of lncRNA in effects of uremia on intestinal TJPs has not been explored.

MiRNA is a non-coding single-stranded RNA with a length of about 20 nucleotides. It inhibits mRNA translation or degradation by binding to the 3 'UTR of mRNA, thereby mediating or silencing gene expression [20]. Previous studies have found that miRNA plays an important role in regulating the body's barrier function. For example, inhibition of miR-155 can significantly inactivate NF-κB signaling pathway and alleviate intestinal inflammation and barrier dysfunction in septic mice [21]. In acute pancreatitis rat model, miR-122 can regulate the expression of Occludin to destroy intestinal barrier function and promote the development of acute pancreatitis [22]. miR-101 disrupted the permeability of the human blood- brain barrier under hypoxic conditions by altered the expression of VE-cadherin and Claudin-5[23]. miR-122-5p exosomes were significantly increased in LPS-induced neutrophils and increased the permeability of brain microvascular endothelial cells by targeting the downstream gene OCLN [24]. In patients with oligospermia, miR-122-5p expression was negatively correlated with OCLN expression [25]. It is suggested that such target relationship may also exist in other organs, tissues or cells. According to bioinformatics analysis (https://starbase.sysu.edu.cn/), miR-122-5p may be a potential target of lncRNA NEAT1. However, whether miR-122-5p is involved in the injury of intestinal epithelial TJP in uremic environment has not been clarified.

In this study, we aimed to investigate the relaitionship of lncRNA NEAT1 and miR-122-5p in uremic toxin-induced intestinal epithelial injury, as well as the underlying molecular mechanisms. The results of the research were to provide new strategy for improving the development of intestinal barrier dysfunction and complications in patients with uremia.

## Materials and methods

### Cell culture and urea treatment

Human embryonic kidney cells (HEK-293T) and human colon cancer cells (Caco-2) were purchased from the Wuhan Procell Life Science and Technology Company. HEK-293T was clutured in high-sugar DMEM medium containing 1% penicillin/streptomycin (Procell, China) and 10% fetal bovine serum (Procell, China). Caco-2 was clutured in MEM medium containing 1% penicillin/streptomycin (Procell, China) and 20% fetal bovine serum (Procell, China). The cells were all incubated at 37°C,5% CO2 condition. Caco-2 cells were treated with different concentrations of urea (0, 42, 72, and 144mg/dl) and incubated for 24 h or 48h. This study was approved in writing by the Ethics Committee of the General Hospital of Ningxia Medical University.

### Lentivirus infection and transfection

The Lentivirus vector for lncRNA NEAT1 (LV-NEAT1) or negative control (LV-NC) were designed and consturuted, from Genechem company (Shanghai, China). LV-NEAT1 was sub-cloned into lentiviral plasmids to infect Caco-2 cells along with lentiviral packaging plasmids. To identify cells stably overexpression of lncRNA NEAT1, the cells were treated with 2μg/mL puromycin for 48h after transfection. miR-122-5p inhibitor, miR-122-5p mimic, inhibitor-NC, mimic-NC were purchased from GenePharma (Shanghai, China). The Caco-2 cell lines were transfected with the above plasmids using Lipofectamine2000(Shanghai, China) followed the manufacturers' instructions.

## Permeability and polarity measurement of Caco-2 monolayer cell model

The success of the monolayer model was evaluated by evaluating the polarity of Caco-2 monolayer model at different time and the permeability of sodium fluorescein. $2 \times 105$ cells were plated on 6.5 mm inserts containing 0.8 μm polyester membranes. Cells were cultured for 21 days in the medium and completely differentiated. An alkaline phosphatase assay (AKP) kit (Nanjing, China) was used following the manufacturer's instructions to evaluate the apical side and basolateral side alkaline phosphatase activity at different time points. Permeability of the monolayer Caco-2 cell measurements were performed by the flux of fluorescein sodium (Sigma, USA). fluorescein sodium (30 mg/L) was added to the apical side of the insets. After 1 h of incubation, the basolateral medium aliquots were collected for the measurement of fluorescence at 490-nm emission wavelengths.

## Cell viability assay

The urea cytotoxicity to Caco-2 was evaluated using a CCK-8 assay (Shanghai, China). Caco-2 cells were seeded in 96-well plates at a density of $2 \times 103$/well and cultured for 24 h, and then treated with different concentrations of urea for 24 h, 48 h, and 72h. Then, added 10 μL CCK-8 solution, incubate for 2 h, and measure the absorbance at 450-nm wavelength with microplate instrument (PerkinElmer, US). Each experiment was carried out three times.

## RT- qPCR analysis

Total RNAs were isolated from cells using TRIzol reagent (Shanghai, China) as the instructions. mRNAs were reverse-transcribed into cDNA by a reverse transcript kit (Takara, Japan). For miRNAs, reverse transcription was performed using a cDNA first chain miRNA synthesis kit (KGIBio, China). SYBR Green PCR Master Mix (Takara, Japan) and fluorescence quantitative PCR miRNA kit (KGIBio, China) were used for qRT-PCR. All responses are analyzed by the CFX Connect Real-time System (BIO-RAD). The relative expression levels were calculated by $2^{-\triangle\triangle Ct}$. The primer sequence is as follows:NEAT1(forward),TGGCATGCTCAGGGCTTCAG(reverse),
TCTCCTTGCCAAGCTTCCTTC;GAPDH(forward),GAAGGTGAAGGTCGGAGTC,(reverse),GAAGATGGTGATGG GATTTC;miR-122-5p(forward),TGGAGTGTGACAATGGTGTTTG,(reverse)AGTGCAGGGTCCGAGGTATT;U6 (forward), AACGAGACGACGACAGAC (reverse), GCAAATTCGTGAAGCGTTCCATA.

## Western blotting

Total protein was extracted from the cells with RIPA lysis buffer (Kaikyl, Shanghai, China), and quantified with BCA assay kit (Kaikyl, Shanghai, China) according to the manufacturers' instructions. Then, the proteins were separated by 6~12% SDS- PAGE and transferred on a PVDF membrane (Millipore, Billerica, MA, USA). After blocking with 5% skimmed milk for 1h at room temperature, the membranes were incubated with ZO-1(1:1000, proteintech, China), Occludin (1:2000, ZSBG-Bio, China), Claudin-1(1:1000, ZSBG-Bio, China), GAPDH (1:10000, ZSBG-Bio, China) overnight at 4°C.

## Immunofluorescence analysis

Cells were cultured to cover slips in a 6-well plate and fixed with 4% paraformaldehyde for 10~15 minutes at room temperature. Afterwards, the cells were permeabilized with 0.3% Triton X-100 for 10 minutes and incubated in a blocking buffer containing 5% BSA for 20 minutes. The primary antibodies such as Occludin (1: 1000), ZO-1 (1:500) and Claudin-1(1:500) were added and incubated with the cells overnight at 4°C. Then with Alexa Fluor®594-labeled secondary antibodies (1: 500) was incubated for 2 h at room temperature and cell nuclei were stained with DAPI. Finally, Con-focal immunofluorescence microscopy (Olympus) was used to observe and collect images.

## Dual- luciferase reporter assay

The lncRNA NEAT1 3'UTR and Occludin 3'UTR were cloned into the pGL3- Basic vector, while mutant NEAT1 3'UTR and Occludin'UTR were cloned into the pGL3 luciferase vector. Then, HK-393T cells were transfected with Occludin 3'UTR, mutant Occludin 3'UTR, NEAT1 UTR, mutant NEAT13'UTR, followed transfected with miR-122-5p mimic or mimic-NC using Lipofectamine 2000 (Shanghai, China). Luciferase activity was then determined by luciferase kit (Meilunbio, China).

## Statistical analysis

All data were expressed as mean±SD deviation (SD). The independent sample t test and one-way ANOVA were used for two or multiple groups. $P < 0.05$ was considered as statistically significant.

## Results

### Urea injury Caco-2 TJPs and down-regulated lncRNA NEAT1 expresion

To construct the Caco-2 intestinal epithelial barrier injury model, the Caco-2 cells were treated with different concentration urea(0、42、72、144mg/dl) at three different time points (24,48 and 72h).According to the results of CCK-8 assay, the viabilities of Caco-2 cells did not show any significant impact on Caco-2 proliferation by the treatment (Fig 1A–1C). Then, the effect of different concentrations of urea on the expression of TJPs ZO-1, Occludin and Claudin-1 was evaluated at 24h and 48h. The results showed that at 24 hours, different concentrations of urea had a damaging effect on ZO-1 in Caco-2 cells ($P < 0.01$), and the Occludin of Caco-2 cells treated with 144mg/dl urea was significantly damaged ($P < 0.01$), while the Claudin-1 was not damaged (Fig 1D–1G). At 48 hours, different concentrations of urea had a damaging effect on ZO-1 and Occludin ($P < 0.01$). Claudin-1 proteins were significantly disrupted($P < 0.01$) (Fig 1I–1K). The results proved that urea could induce the intestinal epithelial barrier injury model of Caco-2 in 144mg/dl at 48h. Then, we profiled the expression of lncRNA NAET1 in urea-treated Caco-2 cells. qRT-RCR analysis was evaluated and the results demonstrated that lncRNA NAET1 expression was significantly down-regulated in Caco-2 cells (Fig 1L).

### Overexpression of lncRNA NEAT1 could repair urea-induced intestinal epithelial barrier injury

To further clarify the potential mechanisms of NEAT1 on intestinal permeability, The LV-NEAT1 and its NC was infected to the Caco-2 cells. Compared with LV-NC, the expression of NEAT1 was significantly up-regulated. After urea treatment, the expression of NEAT1 was significantly decreased (Fig 2A). After Caco-2 monolayer cell models were established, sodium fluorescein flux assays were performed to assess the role of NEAT1 on the permeability of the Caco-2 monolayer cell model (Fig 2B). After urea treatment, the penetration rate of sodium fluorescein is significantly decreased $P < 0.01$). The penetration rate of sodium fluorescein in LV-NEAT1 group was lower than that in LV-NC group and the penetration rate in LV-NEAT1+urea group was lower than that in urea group ($P < 0.01$). As shown in Western Blot, the expression of TJPs ZO-1 and Occludin was significantly increased in LV-NEAT1 group($P < 0.01$), while the Claudin-1 was no significant change($P > 0.05$). The expressions of ZO-1 and Occludin in the LV- NEAT1+urea group were higher than urea treatment group($P < 0.01$) (Fig 2C–2F). Similarly, fluorescence intensity of ZO-1 and Occludin was significantly increased and the Claudin-1 had no obvious change in the LV-NEAT1 group. The fluorescence intensity of ZO-1 and Occludin in the LV-NEAT1+urea group was higher than that in the urea treatment group (Fig. 2G–2I).

### LncRNA NEAT1 acts as a sponge of miR-122-5p

StarBase v2.0 tool was used to forecast the miRNAs that might be targeted to NEAT1. Based on the results, miR-122-5p had binding sites with NEAT1 (Fig 3A). The dual-luciferase reporter experiment results indicated that the luciferase activity of WT-NEAT1 instead of MUT-NEAT1 reporter could be obviously inhibited by miR-122-5p mimics (Fig 3B). Furthermore,

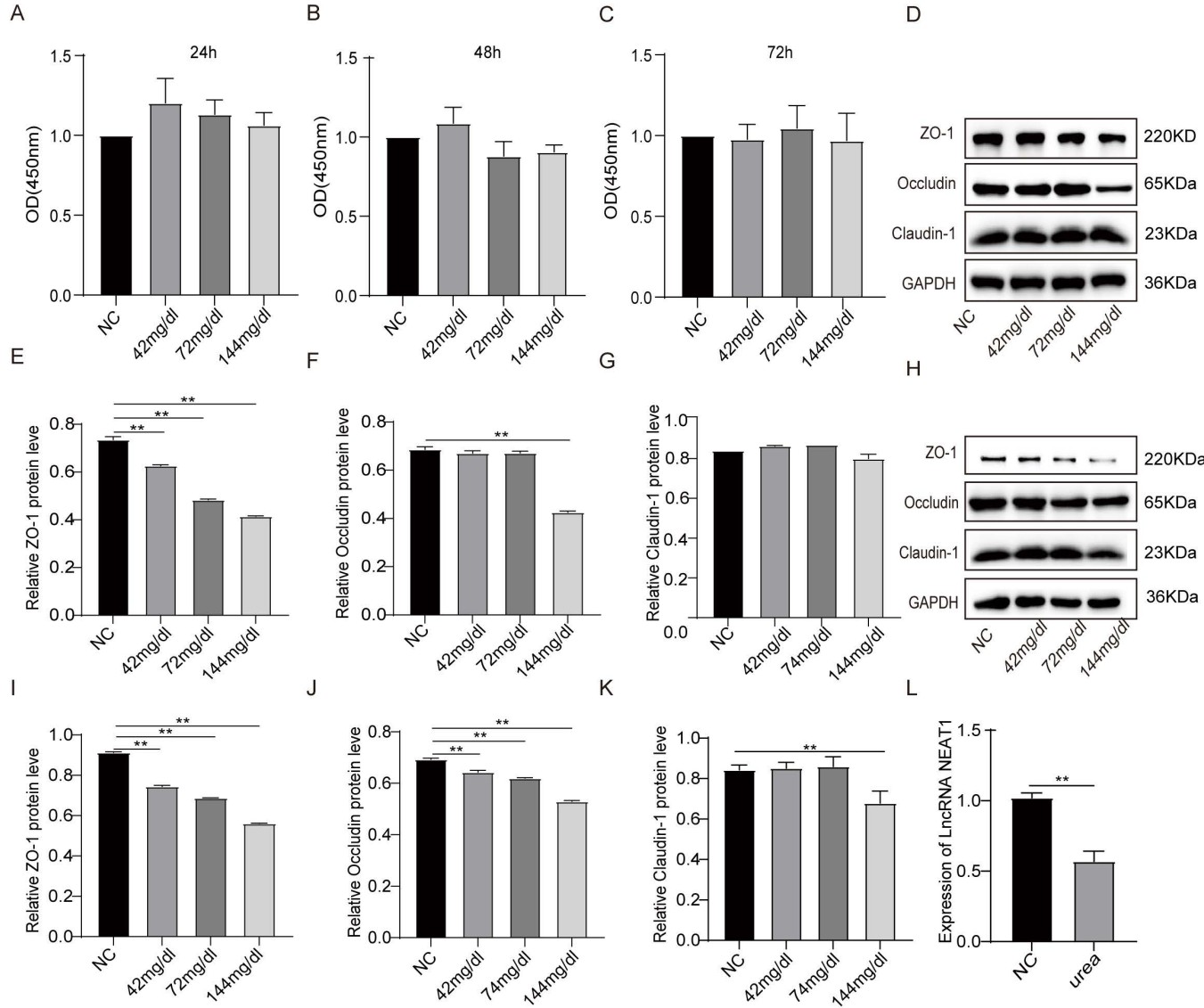

**Fig 1. Urea injury Caco-2 TJPs and down-regulated lncRNA NEAT1 expresion.** (A-C) The effect of different concentrations of urea (42, 72, and 144mg/dl) on cell viability was estimated by CCK8 assay. (D-L) Western blot assay to measure the relative levels of ZO-1、Occludin and Claudin-1 in different concentrations of urea (42, 72, and 144mg/dl) at 24h and 48h. (n = 3, *P < 0.05, **P < 0.01).

144mg/dl urea treatment for 48h significantly promoted the expression of miR-122-5p (Fig 3C). Moreover, NEAT1 overexpresion significantly decreased the expression of miR-122-5p. (Fig 3D).

## Occludin is a target gene of miR-122-5p

MiR-122-5p has been reported to be involved in regulating the permeability of the testicular barrier in mice by targeting Occludin. However, its role in urea-induced intestinal barrier injury has not been elucidated. StarBase v2.0 tool predicted the binding site between miR-122-5p and Occludin (Fig 4A). Dual-luciferase reporter assay showed that miR-122-5p bound to Occludin in a targeted manner (Fig 4B). After transfection with miR-122-5p mimics, The Occludin protein expression was down-regulated. The addition of urea further reduced Occludin expression. When the miR-122-5p inhibitor was

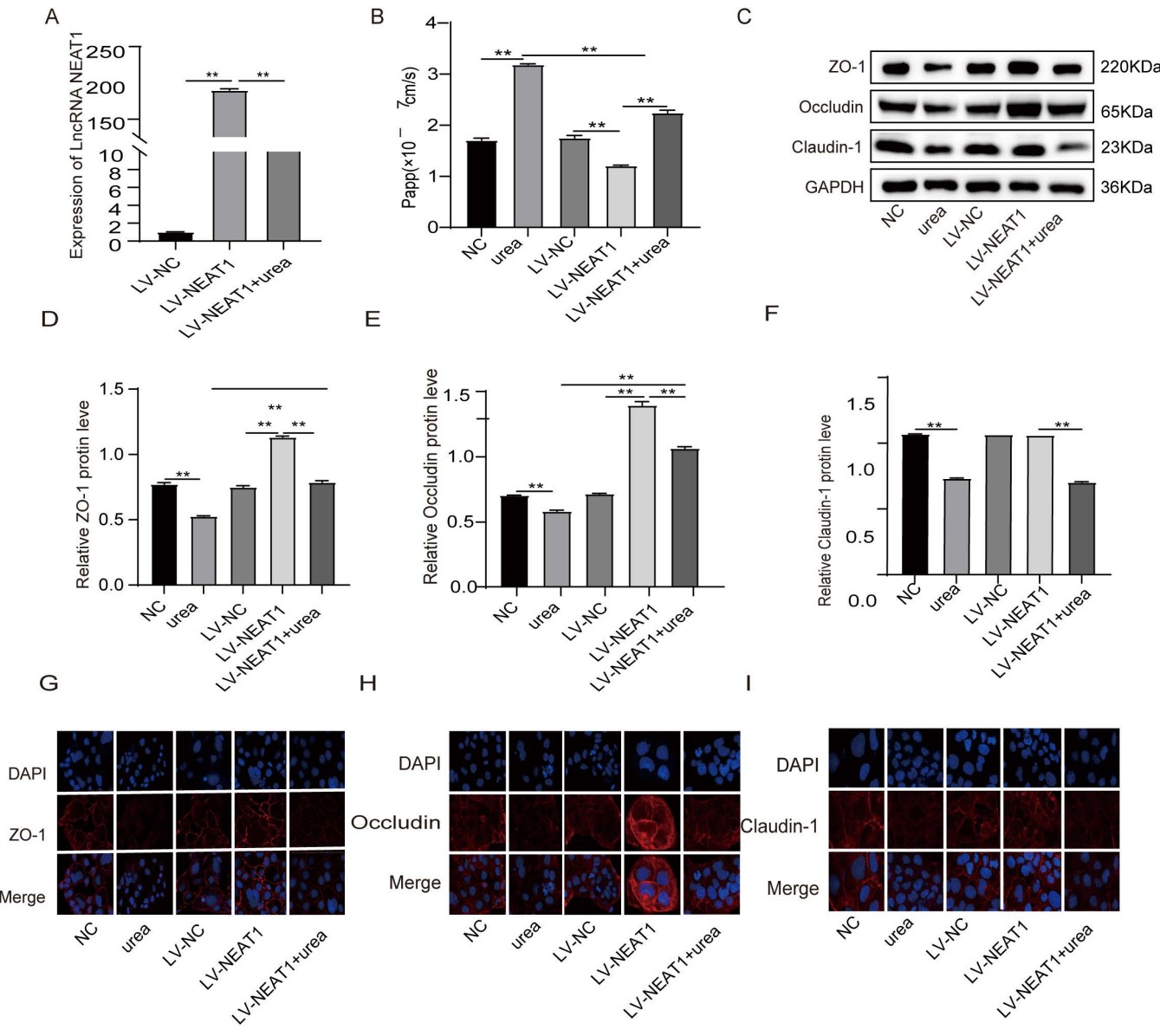

**Fig 2. Overexpression of lncRNA NEAT1 could repair urea-induced intestinal epithelial barrier injury.** (A) After the urea-treated Caco-2 was infected with LV-NEAT1, the expression of NEAT1 was detected by qRT-PCR. (B) Fluorescein sodium transport assay to evaluate the permeability of the monolayer Caco-2 model. (C-F) Western blot assay to measure the relative levels of ZO-1、Occludin and Claudin-1 in these treated Caco-2.(G-I) The expression of ZO-1、Occludin and Claudin-1 was detected by immunofluorescence staining. (n＝3, *P＜0.05, **P＜0.01).

transfected, opposing effects were observed ([Fig 4C–4F](https://doi.org/10.1371/journal.pone.0322989)). these results suggested that miR-122-5p inhibited the expression of Occludin.

## LncRNA NEAT1 functions as a ceRNA for miR-122-5p to regulate Occludin expression in urea-induced injury in Caco-2

To explore whether NEAT1 acted as a sponge of miR-122-5p to regulate Occludin expression, we transfected LV-NEAT1, miR-122-5p mimics, LV-NEAT1＋miR-122-5p into Caco-2 cells. The western blot and immunofluorescence showed

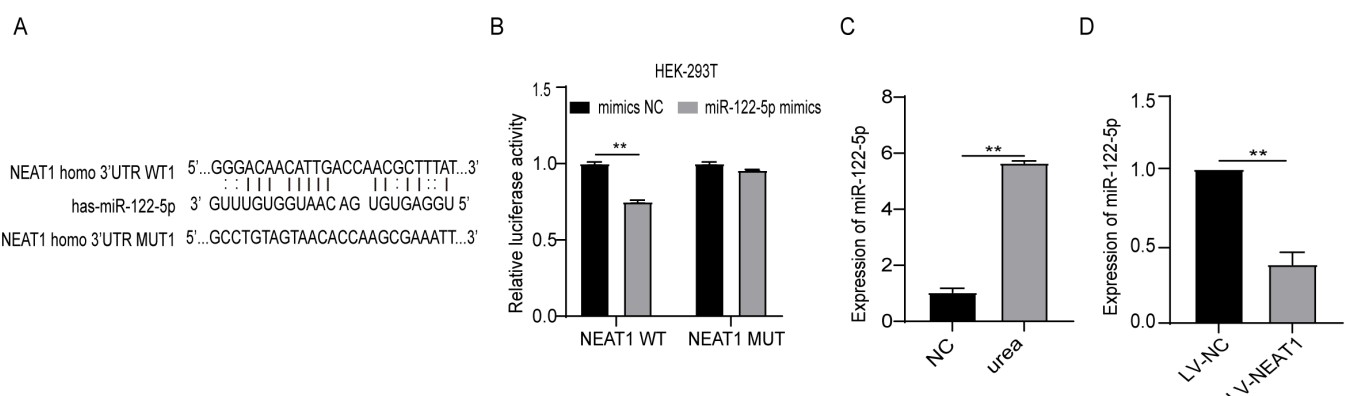

**Fig 3. LncRNA NEAT1 acts as a sponge of miR-122-5p.** (A) StarBase v2.0 predicted binding sites between NEAT1 and miR-122-5p. (B) Relative luciferase activities were performed by dual luciferase reporter assay. (C) The level of miR-122-5p was measured by qRT-PCR in Caco-2 treated with 144mg/dl urea for 48H.(D) After Caco-2 was infected with LV-NEAT1, the expression of miR-122-5p was detected by qRT-PCR. (n = 3, * P < 0.05, **P < 0.01.

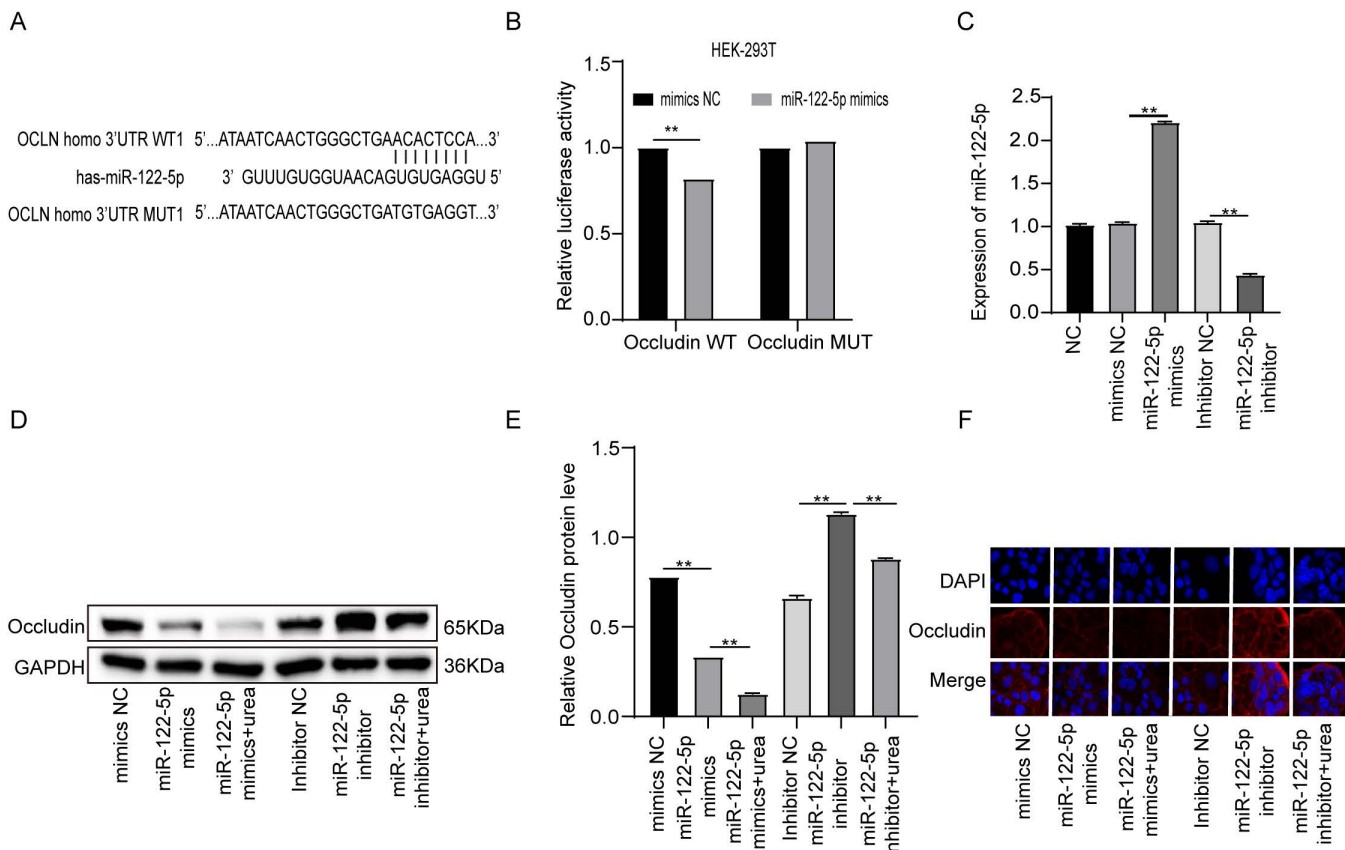

**Fig 4. Occludin is a target gene of miR-122-5p.** (A) StarBase v2.0 predicted binding sites between miR-122-5p and Occludin. (B) Relative luciferase activities were performed by dual luciferase reporter assay. (C) The level of miR-122-5p was measured by qRT-PCR in Caco-2 transfected with miR-122-5p mimics and miR-122-5p inhibitor. (D-E) The relative levels of Occludin measured by Western blot assay. (F) The expression of Occludin was detected by immunofluorescence staining. (n = 3, *P < 0.05, **P < 0.01).

that overexpression of NEAT1 promoted the Occludin expression, while miR-122-5p mimics or urea treatment reduced Occludin expression (Fig 5A–5C). Moreover, overexpression of miR-122-5p significantly inhibited the effect of NEAT1 on the formation of Occludin. These results suggested that NEAT1 positively regulated Occludin expression by targeting miR-122-5p.

## Discussion

CKD patients have obvious intestinal barrier dysfunction, especially ESRD [26]. Intestinal epithelial barrier disruption is both a consequence and a cause of CKD progression. The most important feature of intestinal barrier dysfunction is the significant decrease of TJs, which leads to the increase of intestinal permeability. However, the pathogenesis of CKD-related intestinal dysfunction still lack of understanding. Urea, a representative uremia toxin, increases in the early stages of CKD, and achieves 10 times or more higher than normal in ESRD [27]. In previous studies, Vaziri et al [10] demonstrated that urea significantly increased intestinal permeability and decreased the expression of TJPs including ZO-1, Occludin, and Claudin-1 in T84 cells. In this study, the decreases of above TJPs were also observed in urea-treated Caco2 cells in vitro. However, due to the differences between the cell lines used by us and Vaziri et al, as well as other differences in laboratory conditions, we need to increase the concentration and time of urea to significantly destory these TJPs. Occludin and Claudin-1are adherent transcellular proteins that connect the plasma membranes of adjacent cells, forming a barrier that prevents the diffusion of fluids and solutes [28]. ZO-1, a cytoplasmic protein known as actin binding, serves as an anchor and regulates the organization of the apical junction complex [29]. The reduced distribution and expression of these TJPs will directly lead to the destruction of the TJ structure which leads to microbes, microbial toxins, and bacterial byproducts from the gastrointestinal lumen into the bloodstream, contributing to several systemic consequences, such as systemic inflammation, endotoxemia, and bacterial or endotoxin translocation [30,31]. Therefore, the repair of damaged TJ protein is the key to the treatment of intestinal barrier damage.

It has been well documented that NEAT1 plays pivotal roles in various pathological processes, such as cancer[32], infammation[33]and immune diseases[34]. Previous study suggested that NEAT1 was upregulated in glioma endothelial

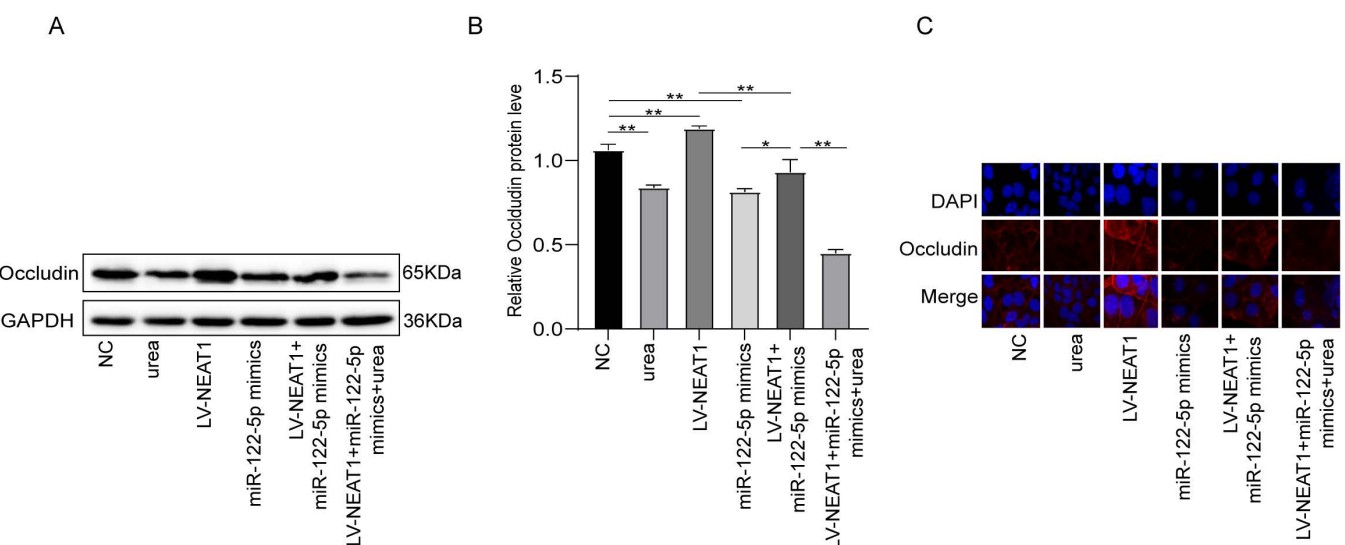

**Fig 5. LncRNA NEAT1 functions as a ceRNA for miR-122-5p to regulate Occludin expression in urea-induced injury in Caco-2.** (A-B) Western blot assay to measure the relative levels of Occludin. (C) The expression of Occludin was detected by immunofluorescence staining. (n = 3, P < 0.05, **P < 0.01).

cells (CECs), knockdown of NEAT can increased blood-tumor barrier permeability through reducing the expression of TJ related proteins including ZO-1, Occludin, Claudin-5. Therefore, we speculated that NEAT1 might also play a role during the progression of urea-induced intestinal epithelial barrier injury. In the present study, our results show that NEAT1 expression was down-regulated in Caco-2 cells exposed to urea. Furthermore, after the Caco-2 cells were cultured in TranswellTM plate for 21 d, the ratio of apical compartment and basolateral compartment AKP was about 4.0, indicating that the polarization of the Caco-2 cell monolayer model was basically completed and the permeability of fluorescein sodium could reach $(1.39 \pm 0.12) \times 10^{-7 \, cm}$ at 1h, indicating that the monolayer Caco-2 cell model was successfully completed [35]. Meanwhile, in this model, overexpression of NEAT1 could decrease the permeability of the Caco-2 monolayer cell and increase the expression of ZO-1and Occludin.

Numerous evidences reported that the close association of the interaction between lncRNAs and miRNAs. It has been reported that lncRNA has miRNA response elements and act as a natural miRNA sponge to reduce binding of endogenous miRNAs to target mRNAs [36,37]. In this study, we screened miR-122-5p as a NEAT1 target binding transcript by bioinformatics analysis combined with dual luciferase reporter assay. Previous study found that miR-122-5p mimics can inhibit the expression of Occludin in the blood-testis barrier and thus increase the permeability of the blood-testis barrier in primary mouse testicular Sertoli cells, while miR-122-5p inhibitors have the opposite result [38]. Furthermore, in infected intestinal porcine epithelial cells (IPEC-J2) with C. perfringens beta2 toxin (CPB2) model, ssc-miR-122–5p was increased in a dose- and time-dependent manner and OCLN was a target of ssc-miR-122–5p. Importantly, ssc-miR-122-5p mimic can lead to cpb2 induced IPEC cell damage further aggravating [39]. In this study, the expression of miR-122-5p and NEAT1 were negatively correlated in the urea-treated group. The miR-122-5p mimics can downregulated Occludin protein levels, whereas the inhibitor of miR-122-5p produced the opposite result. In addition, miR-122-5p mimics could reverse the protective effect of overexpression NEAT1 on Occludin.

However, there were still some shortcomings in our study cannot be neglected. Firstly, due to the long culture period of Caco-2 monolayer cell model, the mimic and inhibitor of miR-122–5-p are transient transfection and cannot be stably expressed in Caco-2 for a long time, so this study did not prove the effect of miR-122–5-p on the permeability of intestinal epithelial model. Secondly, although we have verified that overexpression of lncRNA NEAT1 can also upregulate ZO-1 protein levels, we have not investigated the underlying mechanism. Additionally, the lncRNA NEAT1/miR-122-5p/ Occludin axis was not functionally studied in an in vivo uremic animal model. These limitations are the direction of our future research.

In brief, our study elucidated the function and the molecular mechanism of NEAT1 in uremic induced intestinal barrier injury. It was found that NEAT1 overexpression upregulated the level of Occludin by inhibiting miR-122-5p, thus protecting the intestinal epithelial injury induced by uremic toxins. This study is the first to highlight the importance of the interaction between NEAT1, miR-122-5p and Occludin. In addition, NEAT1/miR-122-5p/Occludin axis, as a promising therapeutic target, will provide new ideas and methods for reducing complications and improving quality of life in patients with uremia.

## Supporting information

**S1 Raw Data.**
(XLSX)

## Author contributions

**Conceptualization:** Meng Han, Libin Wang, Na Tian.

**Data curation:** Lina Fu.

**Formal analysis:** Pathuama Pathuama, Shengnan Zhou.

**Funding acquisition:** Na Tian.

**Investigation:** Pathuama Pathuama.

**Methodology:** Lina Fu.

**Resources:** Jinhai Tian.

**Validation:** Jinhai Tian, Shengnan Zhou, Chen Wang.

**Visualization:** Chen Wang.

**Writing – original draft:** Meng Han.

**Writing – review & editing:** Libin Wang, Na Tian.

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
