## [Decision Letter · Decision Letter 0]

16 Sep 2024

PONE-D-24-22267LncRNA NEAT1 protects uremic toxin-induced intestinal epithelial barrier injury by regulating miR-122-5p/Occludin axisPLOS ONE

Dear Dr. Na,

Thank you for submitting your manuscript to PLOS ONE. After careful consideration, we feel that it has merit but does not fully meet PLOS ONE’s publication criteria as it currently stands. Therefore, we invite you to submit a revised version of the manuscript that addresses the points raised during the review process. Please submit your revised manuscript by Oct 31 2024 11:59PM. If you will need more time than this to complete your revisions, please reply to this message or contact the journal office at plosone@plos.org . Please include the following items when submitting your revised manuscript:

We look forward to receiving your revised manuscript.

Kind regards,

Yash Gupta, Ph.D.

Academic Editor

PLOS ONE

Additional Editor Comments:

The figure quality is not appropriate. Reviewers have raised critical concerns regarding experiment set up and conclusions drawn. Authors need to adress these concerns and add a limitations section to the conclusions.

In the current form this study is not of publication standard.

Reviewers' comments:

Reviewer's Responses to Questions

**Comments to the Author**

1. Is the manuscript technically sound, and do the data support the conclusions?

Reviewer #1: Partly

Reviewer #2: Partly

2. Has the statistical analysis been performed appropriately and rigorously? 

Reviewer #1: Yes

Reviewer #2: Yes

3. Have the authors made all data underlying the findings in their manuscript fully available?

Reviewer #1: Yes

Reviewer #2: Yes

4. Is the manuscript presented in an intelligible fashion and written in standard English?

Reviewer #1: Yes

Reviewer #2: No

5. Review Comments to the Author

Reviewer #1: This study aims to investigate the mechanism of lncRNA Nuclear Enriched Abundant Transcript 1 (NEAT1) in uremic toxin-induced intestinal epithelial barrier injury using Caco-2 and HEK293 cells. The findings demonstrate that lncRNA NEAT1 protects against uremic toxin-induced intestinal epithelial barrier injury by regulating the miR-122-5p/Occludin axis. However, this study has several limitations. It does not provide practical guidance on selecting urea treatment at 144 mg/dL for 48 hours, nor does it identify an optimal model for testing CKD recipient matches. Additionally, it lacks an in-depth analysis of the mechanisms underlying uremic toxin-induced intestinal epithelial injury.

1. The immunofluorescence images presented in this manuscript lack the clarity and resolution necessary for occludin staining (Figures 2G, 2H, 2I, 4F, and 5C).

2. In Figure 1, Western blot images D and H appear to be identical, particularly for ZO-1, claudin-1, and GAPDH. Could you clarify why these images seem to be duplicated?

3. Finally, the conclusion should address the debatable reduction of the tight junction proteins Occludin, ZO-1, and Claudin-1. Additionally, the permeability of sodium fluorescein does not appear to be significantly increased in the model group, which raises questions about the validity of the proposed mechanism involving the miR-122-5p/Occludin axis in supporting the observed injury.

Reviewer #2: This manuscript investigates the role of LncRNA NEAT1 in urea induced Intestinal Epithelial barrier. Authors further explore the interaction between NEAT1 and miRNA-122-5p.

This manuscript requires major revision before it could be accepted for publication. The language is not very easy to comprehend and would require significant change.

Authors have provided complete data that they have discussed in the paper. However, their final conclusion that "NEAT1 acts as a sponge of miRNA-1225p to regulate Occludin expression" need more investigation. Figure5B shows that overexpression of LV-NEAT1 in the presence of miR-122-5p mimic and urea fails to rescue expression of Occludin, which doesnt support their conclusion.

I recommend that authors should evaluate expression of Occludin in a dose-dependent response for both miR-122-5p mimic and urea independently, In the presence of a fixed concentration of other molecule and overexpression of LV-NEAT1.

6. PLOS authors have the option to publish the peer review history of their article (what does this mean? ). If published, this will include your full peer review and any attached files.

**Do you want your identity to be public for this peer review?** For information about this choice, including consent withdrawal, please see our Privacy Policy .

Reviewer #1: No

Reviewer #2: **Yes: ** Shwetank

---

## [Author Response · Author response to Decision Letter 1]

6 Nov 2024

Dear reviewer,

Thank you very much for your comments and professional -advice. These opinions help to improve academic rigor of our article. Based on your suggestion and-request, we have made corrected modifications on the revised manuscript. Furthermore, we would like to show the details as follows

Question1

Please ensure that your manuscript meets PLOS ONE's style requirements, including those for file naming. The PLOS ONE style templates can be found at https://journals.plos.org/plosone/s/file?

id=wjVg/PLOSOne_formatting_sample_main_body.pdfandhttps://journals.plos.org/plosone/s/file?id=ba62/PLOSOne_formatting_sample_title_authors_affiliations.pdf.

The author’s answer:

We have revised our manuscript as requested by PLOS One.

Question2

We note that you have indicated that there are restrictions to data sharing for this study. For studies involving human research participant data or other sensitive data, we encourage authors to share de-identified or anonymized data. However, when data cannot be publicly shared for ethical reasons,we allow authors to make their data sets available upon request. For information on unacceptable data access restrictions, please seehttp://journals.plos.org/plosone/s/data-availability#loc-unacceptable-data-access-restrictions.

The author’s answer:

We have chosen to upload the raw data to PLOS One as supporting information.

Question3

Please include your full ethics statement in the ‘Methods’ section of your manuscript file. In your statement, please include the full name of the IRB or ethics committee who approved or waived your study, as well as whether or not you obtained informed written or verbal consent. If consent was waived for your study, please include this information in your statement as well.

The author’s answer:

We have included the full ethics statement in the Methods section of the manuscript.

Question4

 Please include a separate caption for each figure in your manuscript.

The author’s answer:

We have included a separate caption for each figure in our manuscript.

Question5

PLOS ONE now requires that authors provide the original uncropped and unadjusted images underlying all blot or gel results reported in a submission’s figures or Supporting Information files. This policy and the journal’s other requirements for blot/gel reporting and figure preparation arehttps://journals.plos.org/plosone/s/figures#loc-preparing-figures-from-image-files. When you submit your revised manuscript, please ensure that your figures adhere fully to these guidelines and provide the original underlying images for all blot or gel data reported in your submission. See the following link for instructions on providing the original image data:https://journals.plos.org/plosone/s/figures#loc-original-images-for-blots-and-gels.  

The author’s answer:

We provide raw uncropped and unadjusted images of all blot or gel results in the Supporting Information file.

Additional questions

The figure quality is not appropriate. Reviewers have raised critical concerns regarding experiment set up and conclusions drawn. Authors need to adress these concerns and add a limitations section to the conclusions.In the current form this study is not of publication standard.

The author’s answer:

We have modified the quality of the figure.

Reviewer #1: This study aims to investigate the mechanism of lncRNA Nuclear Enriched Abundant Transcript 1 (NEAT1) in uremic toxin-induced intestinal epithelial barrier injury using Caco-2 and HEK293 cells. The findings demonstrate that lncRNA NEAT1 protects against uremic toxin-induced intestinal epithelial barrier injury by regulating the miR-122-5p/Occludin axis. However, this study has several limitations. It does not provide practical guidance on selecting urea treatment at 144 mg/dL for 48 hours, nor does it identify an optimal model for testing CKD recipient matches. Additionally, it lacks an in-depth analysis of the mechanisms underlying uremic toxin-induced intestinal epithelial injury.

1. The immunofluorescence images presented in this manuscript lack the clarity and resolution necessary for occludin staining (Figures 2G, 2H, 2I, 4F, and 5C).

2. In Figure 1, Western blot images D and H appear to be identical, particularly for ZO-1, claudin-1, and GAPDH. Could you clarify why these images seem to be duplicated?

3. Finally, the conclusion should address the debatable reduction of the tight junction proteins Occludin, ZO-1, and Claudin-1. Additionally, the permeability of sodium fluorescein does not appear to be significantly increased in the model group, which raises questions about the validity of the proposed mechanism involving the miR-122-5p/Occludin axis in supporting the observed injury.

---

## [Decision Letter · Decision Letter 1]

19 Jan 2025

PONE-D-24-22267R1LncRNA NEAT1 protects uremic toxin-induced intestinal epithelial barrier injury by regulating miR-122-5p/Occludin axisPLOS ONE

Dear Dr. Na,

Thank you for submitting your manuscript to PLOS ONE. After careful consideration, we feel that it has merit but does not fully meet PLOS ONE’s publication criteria as it currently stands. Therefore, we invite you to submit a revised version of the manuscript that addresses the points raised during the review process.

We look forward to receiving your revised manuscript.

Kind regards,

Yash Gupta, Ph.D.

Academic Editor

PLOS ONE

Journal Requirements:

Reviewers' comments:

Reviewer's Responses to Questions

**Comments to the Author**

1. If the authors have adequately addressed your comments raised in a previous round of review and you feel that this manuscript is now acceptable for publication, you may indicate that here to bypass the “Comments to the Author” section, enter your conflict of interest statement in the “Confidential to Editor” section, and submit your "Accept" recommendation.

Reviewer #1: All comments have been addressed

Reviewer #3: All comments have been addressed

2. Is the manuscript technically sound, and do the data support the conclusions?

Reviewer #1: Yes

Reviewer #3: Yes

3. Has the statistical analysis been performed appropriately and rigorously? 

Reviewer #1: I Don't Know

Reviewer #3: I Don't Know

4. Have the authors made all data underlying the findings in their manuscript fully available?

Reviewer #1: Yes

Reviewer #3: Yes

5. Is the manuscript presented in an intelligible fashion and written in standard English?

Reviewer #1: Yes

Reviewer #3: Yes

6. Review Comments to the Author

Reviewer #1: I still have some concerns regarding the manuscript. For the Western blots, please ensure the molecular weight (kDa) is specified for each protein. Additionally, clarify how the Western blot bands were quantified, and ensure this is clearly described. Specify the number of experiments conducted. Lastly, there are additional references missing in the manuscript that should be included in the final version.

Reviewer #3: The author has carefully considered the comments and revised the manuscript to enhance its scientific validity and transparency.

7. PLOS authors have the option to publish the peer review history of their article (what does this mean? ). If published, this will include your full peer review and any attached files.

**Do you want your identity to be public for this peer review?** For information about this choice, including consent withdrawal, please see our Privacy Policy .

Reviewer #1: No

Reviewer #3: No

---

## [Author Response · Author response to Decision Letter 2]

18 Feb 2025

Dear reviewer,

Thank you very much for your comments and professional -advice. These opinions help to improve academic rigor of our article. Based on your suggestion and-request, we have made corrected modifications on the revised manuscript. Furthermore, we would like to show the details as follows

As some references were retracted, we have replaced them to ensure the accuracy of the article, which has been standardized in the article.

Reviewer #1: I still have some concerns regarding the manuscript. For the Western blots, please ensure the molecular weight (kDa) is specified for each protein. Additionally, clarify how the Western blot bands were quantified, and ensure this is clearly described. Specify the number of experiments conducted. Lastly, there are additional references missing in the manuscript that should be included in the final version.

My answer:Thanks for your suggestion. We have annotated the molecular weight of each protein in the figures of the article.The gray value of the target protein bands was quantitatively analyzed by imageJ, and each experiment was repeated 3 times to determine the expression level of the target protein.AdditionIy, we have refined and revised the references for completeness and accuracy.

Reviewer #3: The author has carefully considered the comments and revised the manuscript to enhance its scientific validity and transparency.

My answer:Thank you for your review. We tried our best to improve the manuscript and made some changes to the manuscript. These changes will not influence the content and framework of the paper.

---

## [Editor Report · Decision Letter 2]

1 Apr 2025

LncRNA NEAT1 protects uremic toxin-induced intestinal epithelial barrier injury by regulating miR-122-5p/Occludin axis

PONE-D-24-22267R2

Dear Dr. Na,

We’re pleased to inform you that your manuscript has been judged scientifically suitable for publication and will be formally accepted for publication once it meets all outstanding technical requirements.

Kind regards,

Yash Gupta, Ph.D.

Academic Editor

PLOS ONE
---

## [Editor Report · Acceptance letter]

PONE-D-24-22267R2

PLOS ONE

Dear Dr. Tian,

I'm pleased to inform you that your manuscript has been deemed suitable for publication in PLOS ONE. Congratulations! Your manuscript is now being handed over to our production team.

Kind regards,

on behalf of

Dr. Yash Gupta

Academic Editor

PLOS ONE